

# Designing a network of critical zone observatories

# to explore the living skin of the terrestrial Earth

Susan L. Brantley, Earth and Environmental Systems Institute, Dept of Geosciences, Pennsylvania State University, Univ. Pk, PA 16802

William H. McDowell, Department of Natural Resources and the Environment, University of New Hampshire, Durham, NH, 03823

William E. Dietrich, Department of Earth and Planetary Science, UC Berkeley, CA 94720

Timothy S. White, Earth and Environmental Systems Institute, Dept of Geosciences, Pennsylvania State University, Univ. Pk, PA 16802

Praveen Kumar, Department of Civil and Environmental Engineering, University of Illinois, Urbana, Illinois 61801

Suzanne Anderson, Department of Geography, Institute of Arctic and Alpine Research (INSTAAR), University of Coloardo, Campus Box 450, Boulder, CO 80309-0450

Jon Chorover, Department of Soil, Water and Environmental Science, University of Arizona, Tucson, AZ 85750

Kathleen Ann Lohse, Department of Biological Sciences, Idaho State University, Pocatello, ID 83209

Roger C. Bales, Sierra Nevada Research Institute, University of California, Merced 94530

Daniel deB. Richter, Nicholas School of the Environment, Duke University, Durham, North Carolina 27708

Gordon Grant, Pacific Northwest Research Station, USDA Forest Service, Corvallis, OR 97331

Jérôme Gaillardet, Institut de Physique du Globe de Paris, Sorbonne Paris Cité, CNRS, Paris

**Abstract.** The critical zone (CZ), the dynamic living skin of the Earth, extends from the top of
the vegetative canopy through the soil and down to fresh bedrock and the bottom of
groundwater.  All humans live in and depend on the critical zone. This zone has three co-
evolving surfaces: the top of the vegetative canopy, the ground surface, and a deep subsurface
below which Earth's materials are unweathered.  The network of nine critical zone observatories
supported by the U.S. National Science Foundation has made advances in three broad areas of
critical zone research relating to the co-evolving surfaces.  First, monitoring has revealed how
natural and anthropogenic inputs at the vegetation canopy and ground surface cause subsurface
responses in water, regolith structure, minerals, and biotic activity to considerable depths. This
response, in turn, impacts above-ground biota and climate. Second, drilling and geophysical
imaging now reveal how the deep subsurface of the CZ varies across landscapes, which in turn
influences above-ground ecosystems. Third, several new mechanistic models now provide
quantitative predictions of the spatial structure of the subsurface of the CZ.
Many countries fund critical zone observatories (CZOs) to measure the fluxes of solutes, water,
energy, gases, and sediments in the CZ and some relate these observations to the histories of
those fluxes recorded in landforms, biota, soils, sediments, and rocks. Each U.S. observatory has
succeeded in i) synthesizing research across disciplines into convergent approaches; ii) providing
long-term measurements to compare across sites; iii) testing and developing models; iv)
collecting and measuring baseline data for comparison to catastrophic events; v) stimulating new
process-based hypotheses; vi) catalyzing development of new techniques and instrumentation;
vii) informing the public about the CZ; viii) mentoring students and teaching about emerging
multi-disciplinary CZ science; and ix) discovering new insights about the CZ. Many of these
activities can only be accomplished with observatories. Here we review the CZO enterprise in
the US and identify how such observatories could operate in the future as a network designed to
generate critical scientific insights. Specifically, we recognize the need for the network to study
network-level questions, expand the environments under investigation, accommodate both
hypothesis testing and monitoring, and involve more stakeholders. We propose a driving
question for future CZ science and a "hubs-and-campaigns" model to address that question and
target the CZ as one unit. Only with such integrative efforts will we learn to steward the life-
sustaining critical zone now and into the future.


## 1 Introduction

Humans live at Earth's surface – a surface that changes at timescales ranging from milliseconds to millions of years. Understanding how to sustain a growing human population on this changing substrate while simultaneously sustaining diverse ecosystems is a grand challenge for scientists and decision makers (Millenium Ecosystem Assessment Board, 2005; Easterling, 2007). In recognition of the critical nature of Earth's surface, the United States (U.S.) National Research Council defined the zone from the upper vegetative canopy through ground water as the "critical zone" and identified study of this "CZ" as one of the *Basic Research Opportunities in the Earth Sciences* (U.S. National Research Council Committee on Basic Research Opportunities in the Earth Sciences, 2001).

While the critical zone was defined in 2001, only recently has it been recognized as a distinct co-evolving entity driven by physical, chemical, and biological processes that sustain life. To tackle the critical zone for the first time as an entity – rather than study pieces of it separately – is a paradigm shift in science (Fig. 1). Currently, understanding this zone requires researchers drawn from many traditional disciplines including geology, hydrology, climate science, ecology, soil science, geochemistry, geomorphology, and social science to work in collaboration. In the future, it will be pursued not only by disciplinary researchers but also by new scientists trained to cross disciplines. Critical-zone science uses a wide range of measurements as the foundation for advances in understanding and prediction. Scientists quantify fluxes of solutes, water, energy, gases, and sediments (SWEGS) as they are today and then compare them to the histories and impacts of those fluxes recorded over geological time in landforms, regolith structure, soils, and sediments. Critical zone scientists also relate these fluxes to natural and anthropogenic drivers as well as to the structure of the critical zone, including biota, soil, and regolith. In this way, models are developed that can scale CZ properties across the landscape and project the CZ changes across time into the future. From this endeavor has emerged the concept of critical zone science, a new paradigm shift that has been adopted around the world to investigate Earth's surface from canopy to bedrock in its entirety as one integrated unit.

CZ science typically has these attributes: i) it targets Earth's surface from canopy through groundwater; ii) it encompasses timescales from milliseconds (or less) to millions of years; iii) it incorporates insights from all relevant disciplines. Some have described CZ science as an attempt to put more geology into watershed science, or the study of the interaction of rocks and ecosystems. Each of those descriptors falls short of the full complexity of understanding the CZ as an entity with its own identity.

In the U.S., observatories to study the critical zone have been established by the National Science Foundation (NSF). The physical scope of these US CZOs varies, as some are defined by watershed boundaries, some by land use, others by the range of climate conditions, and still

others by contrasts in lithology or geomorphic history.  The common elements are: 1) the focus
on the entire above- and below-ground critical zone and its fluxes, 2) documentation of CZ
structure, 3) mechanistic process studies, and 4) analysis of the history of the landscape that gave
rise to its current structure.  This last feature is a crucial aspect of CZOs that distinguishes them
from previous studies that typically do not consider "deep time" (i.e., geologic time) and often
fail to document the critical zone below the upper soil.  Previous papers have described how
researchers have grappled with the establishment of a measurement design at a specific CZO,
with the overall data needs of CZOs, and with a modeling framework that might be used to
tackle the complexity of timescales under consideration (Horsbaugh et al., 2008; Duffy et al.,
2014; Niu et al., 2014; Brantley et al., 2016).
Over the last decade of study at individual CZOs, cross-CZO science has emerged and
begun to unite the observatories into a CZO network with the capacity to test hypotheses across a
larger parameter space than can be represented by any single CZO.  As a result, network-level
science has begun to emerge and provide opportunities that were not possible in the past. In this
paper, we review the evolution of CZ science as an interdisciplinary "experiment." We take
stock of successes and weaknesses. The goal of the paper is to strategize about how to design a
better CZO network to maximize future opportunities in CZ science at the levels of the
individual observer, the observatory, the network and the broader Earth surface science
community.  Specifically, we address the broad question: what programs and infrastructure are
needed by the community to understand the CZ of the future? The review is focused on activities
and results of this endeavor to date in the U.S. We acknowledge the many other observatory
networks around the world (http://www.czen.org/site_seeker) and invite future papers on how
those other networks are constituted and how the various observatory systems can work together.
One way to review the U.S. program to date is to summarize performance through
quantitative metrics. As manifested today, the program funds nine CZOs situated across a range
of landscapes (Fig.  2). In addition, interdisciplinary field observatories that host critical-zone
science involve thousands of interdisciplinary investigators in more than 25 nations (Fig.  3)
(Giardino and Houser, 2015).  Other metrics further highlight how CZ science has energized
people and approaches in the U.S. and abroad (Tables 1, 2, 3, 4).  Indeed, the term "critical zone"
has been used in 925 papers as of June 2017, in title, abstract, or keyword as recorded in Web of
Science.  The term has even entered the realm of geopolitics (Latour, 2014) and been defined in
scientific dictionaries (White and Sharkey, 2016). This is remarkable given that critical-zone
science as an idea only dates from 2001.  Less quantifiable, however, is the impact of the idea of
the CZ: for example, one country hosting one of the longest-functioning observatory networks in
the world (France) is now specifically promoting communication among disciplines and sites.
Researchers within that network have pointed to the internationally-emerging paradigm of the
CZ as a driver for this new level of communication.  Thus, quantitative metrics such as those in
Table 1 do not fully illustrate the way CZ science has impacted science. In the rest of the paper
we therefore discuss the evolution of observatories in environmental science, other mechanisms

for studying the CZ, the history of the CZO program, the nine roles of CZOs, and CZO measurements and models. We finish by discussing the strengths and weaknesses of the CZO approach to date and show how network-level understanding is starting to emerge. In the last section we consider an overarching question to drive future CZ science, and we propose a new topology for the CZO network.

## 2  Historical context for environmental observatories and networks

It is useful to place the CZ enterprise broadly into the context of environmental science. The differentiated scientific disciplines largely did not yet exist until the 1900s, and the earliest natural scientists therefore tended to be multi-disciplinary (e.g. Forbes, 1887; CFIR CSEPP, 2005; Warkentin, 2006; Berner, 2012; Riebe et al., 2016). These researchers early on began to articulate the need for place-based, integrative science. This has led to a rich history of diverse, place-based, experimental, and long-term observation sites in the U.S. and world.

Many of these observatories were established to investigate the impacts of specific land use activities. One of the first was the Rothamsted Experimental Station in Harpenden, England, founded in 1843 to study the effects of fertilizers on crops. Not until 1910 did the concept of using paired watersheds to investigate the hydrologic and geomorphic impacts of land use treatments within the U.S. begin when a pair of small instrumented catchments were instituted for monitoring to evaluate changes in stream flow and sediment yield at Watson Wheel Gap, Colorado (Van Haveren, 1988).

As human population and land use increased, researchers began to compare pristine sites to human-impacted sites, and began emphasizing the need for long-term measurements (Leopold, 1962). The U.S. Geological Survey thus developed a hydrologic benchmark network (HBN) of 57 basins (Cobb and Biesecker, 1971) to make long-term hydrologic measurements. The mandate of the HBN was expanded in 2011 to include long-term observations not only of stream flow and water quality but also of soil chemistry and aquatic ecology (McHale et al., 2014). Thirty-seven hydrologic benchmark watersheds are still maintained (Mast, 2013), but the original vision to also characterize vegetation and geology has remained unfulfilled.

The paired-watershed approach pioneered in Watson Wheel Gap, Colorado, was replicated much later in many other locations in the U.S., including Coweeta Hydrologic Laboratory (North Carolina), Hubbard Brook (New Hampshire), Reynolds Creek Experimental Watershed (Idaho), and the H.J. Andrews Experimental Forest (Oregon) among others. These pioneer sites, in turn, led to the establishment of over 70 Experimental Forests and Rangelands as sites that focus on fundamental and applied questions spanning hydrology, silviculture, soil science, and climate research (Lugo et al., 2006).

In the early 1980s, as academic scientists pointed out the difficulties of answering questions about long-term natural processes given the realities of short-term funding (Bormann and Likens, 1979; Callahan, 1984), the U.S. National Science Foundation Directorate of Biological Sciences created the Long-Term Ecological Research (LTER) program to carry out observation-based research across a network of sites that spanned the major biotic regions of

North America. These efforts were aided by early work of the U.S. Forest Service to understand
watershed hydrology (Swank and Crossley, 1988). The LTER effort was initiated predominantly
by ecosystem ecologists asking questions about organisms with long life cycles, including how
they are maintained by natural water and nutrient fluxes in the face of acute environmental
changes that are long-term as well as episodic (Dodds and al., 2012). Although not exclusively
based on the study of watersheds as pioneered in the late 1960s (Bormann and Likens 1967),
many LTERs follow a model that emphasizes the study of energy, water, and material flows
through a watershed (Hynes 1975). As of 2017, the LTER network contains 28 LTERs and is
beginning to emphasize the need to incorporate social science (LTER Network Office, 2011).
A big step was taken in 1991 toward incorporating geology into these largely water-land
use-ecology observatories when the U.S. Geological Survey inaugurated the Water, Energy and
Biogeochemical Budgets (WEBB) program. WEBB targeted interactions among water, energy,
and biogeochemical fluxes in five sites chosen at least partly on the basis of their inherent
geological characteristics and relatively pristine condition.
Then, in 2008, another long-term research program was envisioned by agricultural
researchers (Robertson et al. 2008). This vision resulted in the establishment of the Long-Term
Agroecosystem Research (LTAR) program in 2011. This network, today including 18 LTARs,
promotes long-term agricultural research facilities, experiments, and watershed-based studies
focused on sustaining agriculture and increasing crop yields under changing climate conditions
while minimizing or reversing any adverse environmental impacts
(http://www.tucson.ars.ag.gov/ltar/).
The most recent addition to the development of observatories in the U.S. is the National
Ecological Observatory Network (NEON). NEON is a U.S.-wide, distributed observatory that
aims to understand and forecast the impacts of climate change, land use, and invasive species on
ecology and ecosystem fluxes by providing a research platform for investigator-initiated sensors,
observations and experiments that can provide consistent, continental, long-term, multi-scaled
data (Loescher et al., 2017). NEON has 84 sites across the U.S. Like LTER, NEON is a program
funded by the Directorate of Biological Sciences at NSF to study ecological change (Golz et al.,
202 2016).

As in the U.S., international observatory networks have also grown, many for
substantially the same reasons that they grew in the U.S. – to study land use, water, and ecology.
In at least one country (France), a network (OZCAR: Observatoires de la Zone Critique:
applications et recherche) emphasizes individual disciplines at each observatory and provides as
much as 50 years' worth of data to enable research in some sites. The long-term, place-based
ecological research that was pioneered by the LTER network in the U.SA. has also been adopted
by the broader international community in the International LTER (ILTER) network (Vanderbilt
and Gaiser, 2017). Today, the European Commission is promoting an approach to develop a
European Research Infrastructure in the form of a network associating CZOs, LTERs and
LTSERs. Here, LTSER stands for Long Term Socio-Ecological Research, i.e., a network that
also incorporates questions from social science. Indeed, many of the European countries
maintain strong observatory infrastructures that are much more tightly linked with local
stakeholders than observatories in the U.S. This may result from the lack of truly "natural"
territories in Europe, given the long history of development on the continent but also the
willingness to co-construct research questions with stakeholders to build a sustainable
environmental future.

## 3  Non-observatory approaches used to study the CZ

Just as observatory science was beginning with the Watson Wheel Gap observatory in the
early 1900s,  scientists also began to focus on portions of the Earth system that could be
understood in a reductionist sense (Riebe et al., 2016).  Eventually, small-grant funding to single
investigators or small teams became the dominant mechanism to fund research to explore
questions about the CZ. This targeted approach further emphasized reductionism and served to
grow the individual disciplines of geochemistry, geobiology, geomorphology, hydrology, soil
science, ecology, meteorology, and others. Disciplinary growth in turn allowed relatively defined
"monodisciplinary" paradigms to mature and led to the proliferation of disciplinary journals. For
example, Web of Science currently indexes 225 journals in the fields of environmental sciences,
184 in geosciences, and 150 in ecology, with some journals cross-reported in more than one
category.
Through smaller funded projects, many different types of measurements were made.
However, the measurements were completed at different sites and integration of observations
into models was difficult to impossible. Advances in studying Earth's surface tended to be
uneven because different sites were targeted and coordination among disciplinary approaches
was lacking. Such fragmentation accentuated the need for observatories. Other mechanisms were
also needed, however, as questions about environmental impacts on human health grew in the
U.S. throughout the 1970s (CFIR CSEPP, 2005). Funding agencies began seeking teams of
researchers to pursue campaigns –  concerted, multi-investigator, multi-year projects – targeting
focused hypotheses about landforms, soils, water, biota, in addition to human health. Such
campaigns culminated in global efforts such as the Millenium Ecosystems Assessment
(Millenium Ecosystem Assessment Board, 2005) and the International Geosphere-Biosphere
Program (IGBP). The latter initiative engaged 10,000 scientists from more than 20 disciplines
and 80 countries (CFIR CSEPP, 2005; Millenium Ecosystem Assessment Board, 2005).
Eventually, another type of funding mechanism to study the CZ emerged in the U.S.
alongside observatory, single-investigator, and campaign-style science. Specifically, centers of
excellence were funded to draw together scientists into institutions to focus on specific problems
or approaches. One impetus for this was the inauguration in 1987 of the NSF Science and
Technology Center program. This effort eventually supported two centers of special relevance to
critical zone research:  SAHRA (Sustainability of Semi-Arid Hydrology and Riparian Areas) and
NCED (National Center for Earth-surface Dynamics). NCED (2002 to 2012) focused on
developing a quantitative, predictive Earth-surface science by integrating geomorphology,
ecology, hydrology, sedimentary geology, engineering, social sciences, and geochemistry by
combining field, experiment, and computational approaches. NCED and its reincarnation as
NCED2 after 2012 both emphasize predictive Earth-surface science. A similarly ambitious
institution, the National Center for Ecological Analysis and Synthesis (NCEAS), was established
in 1995 as the first national synthesis center for ecology. Neither of these centers focused on the
CZ as one single entity.
Other examples of institutionalized centers of excellence also were important in
developing aspects of CZ science. For example, the Community Surface Dynamics Modeling
System (CSDMS; http://csdms.colorado.edu/wiki/Main_Page) is building and promoting a
library of models for various Earth surface processes by supporting a broad community of
modelers. The National Center for Airborne Laser Mapping (NCALM, established in 2003)
provides research-quality airborne light detection and ranging (LiDAR) observations to the
community. Another example is the Consortium of Universities for the Advancement of
Hydrologic Science, Inc. (CUAHSI; https://www.cuahsi.org/), which aims to advance hydrologic
sciences broadly across the U.S. and its member universities. Other centers of excellence have
been established to promote use of instrumentation such as the NSF-funded Purdue Rare Isotope
Measurement Laboratory (PRIME Lab), a dedicated research and service facility for accelerator
mass spectrometry (AMS) including measurement and interpretation of cosmogenic isotopes.

## 270    4  The CZO program

Even with this variety of funding mechanisms for Earth and environmental science, no
concerted nationwide effort emerged to tackle questions and to train students to target the CZ as
one entity, incorporating the deep geological underpinnings and long-timescale perspectives. As
a result, the environmental science that developed often had to rely on statistical approaches to
explain variability instead of developing more fundamental explanations based on underlying
geological heterogeneity and its origins. Recognizing the need to emphasize the geological
underpinnings of place-based science in the late 2000s, researchers within the water, soil,
geochemistry, and geomorphology communities began articulating a need for integrated science
across the entire zone from canopy to bedrock to incorporate the full significance of the
underlying geology (Anderson, 2004; Brantley et al., 2006; Chorover et al., 2007; U.S.
Committee on Integrated Observations for Hydrologic and Related Sciences, 2008; U.S. Steering
Committee for Frontiers in Soil Science, 2009; U.S. National Research Council, 2010; Banwart
et al., 2011; Committee on New Research Opportunities in the Earth Sciences at the National
Science Foundation, 2012; White and Sharkey, 2016).
Eventually the need to study the CZ as one integrated entity resulted in the NSF program
establishing the Critical Zone Observatory program in 2007 (White et al., 2015). In this initial
phase, three CZOs were funded (Anderson et al., 2008). Two years later, three more CZOs were
funded. By 2013 this number had grown to nine observatories supported through a competitive
selection process. In addition to the expansion of sites in 2013, a CZO National Office (NO) was
established by NSF in 2014 through a competitive process, with the intent of providing the CZO
Network with an administrative structure for furthering coordination (White et al., 2015). The
number of CZOs has remained stable through 2017.
Inauguration of the CZO program implicitly defined the term "critical zone observatory"
to be distinct within the long history of observatories in the US and abroad as an observatory that
promotes study of the entire CZ as one entity. As implemented today, CZOs are sites or closely
connected sets of sites with no required size or specified range of conditions.  In fact, the
physical scope of a CZO is set only by the fundamental questions driving the establishment of
the observatory.  A fundamental characteristic of a CZO is that it is able to operate over a long
enough period to quantify controlling mechanisms thoroughly and to capture temporal trends that
reveal how the critical zone operates.  Two more characteristics of a CZO are that it is amenable
to study by many disciplines and that it integrates understanding of long- and short-timescale
phenomena.  Finally, each CZO operates as an adaptive and agile hypothesis-testing machine,
not simply a monitoring program.  As CZOs developed in the U.S., they began to play nine
important roles within the environmental scientific endeavor. These are described in the next
section.

## 5  The nine emergent roles of CZOs

Here the nine important roles of an observatory are described with examples of scientific
results from across the CZO network today.
First, CZOs act as *synthesizers of interdisciplinary research into convergent approaches*
at one specific site that lead to novel understanding and ultimately result in more deeply-
informed generalized and predictive understanding (Rasmussen et al., 2011a).  In other words,
observatories induce scientists from different disciplines to make measurements using different
disciplinary approaches at the same location instead of making them at disparate sites, driving
cross-disciplinary understanding in describing CZ function (Hynek et al., 2016; Sullivan et al.,
2016; Yan et al., 2017; Chen et al., 2017 in press). At first, much of the synthesis crossed only
two disciplines at a time:  for example, several papers emphasized how geomorphological
concepts related to erosion must be incorporated to understand chemical weathering, and vice
versa (Rempe and Dietrich, 2014; Riebe et al., 2016). Likewise, researchers have related tree
roots to water cycling (Vrettas and Fung, 2015). Now, researchers are targeting multi-disciplinary
aspects of CZ entities.  For example, at the Calhoun CZO, where the South Carolina landscape
was severely eroded by cotton farming, logistic regression models treat market and policy
conditions in the context of topographic characteristics (Coughlan et al. 2017). By fostering

measurements from all disciplines in centralized places, CZOs are discovering not only how to cross disciplines but how individual disciplines can converge.

Second, CZOs provide *stable platforms for long-term measurements* (Table 2).  Some datasets synthesized by CZOs are now available for decades or several decades. For example, the Reynolds Creek CZO recently published 31 years of hourly data that are spatially distributed at 10 m resolution for air temperature, humidity, and precipitation (Kormos et al., 2016) and a 10-year data set that spans the rain-snow transition (Enslin et al., 2016). Similarly, decreasing trends in water and energy influx in the Jemez CZO over the past 30 years were recently related to CZ structure (Zapata-Rios et al., 2016).  Major changes in soil biogeochemistry have been documented by Calhoun CZO researchers over 50-years of reforestation in fields cultivated for cotton (Mobley et al., 2015). That CZO also spearheads an effort to recover archived data from three eroded watersheds that were farmed from the late 1940s to 1962 – as well as to re-instrument the catchments.  Many other multi-year measurements common to all CZOs enable hypothesis-testing.  For example, characterization of dissolved organic matter (DOM) measured with similar methodology across five CZOs revealed a strong role for CZ structure in setting the origin, composition and fate of DOM in streams (Miller et al., 2016).  In another example, a coordinated effort emerged to measure and understand the relationships among solute concentrations and water discharge in streams (e.g.  Kirchner, 2003; Godsey et al., 2009). A special issue on the topic (Chorover et al., 2017, in press) is pointing the way toward the use of knowledge of subsurface structure to explain concentration-discharge behavior *a priori*.

Third, CZOs act as *a stimulus and test-bed for modelling and prediction*. Modeling the CZ is a unique challenge in that models must address the coupling across time scales from seconds to millennia (Table 3).  To tackle this challenge, CZOs are both adapting existing models and developing new models. For example, one CZO is developing a hierarchy of modules to describe processes that occur over seconds to millennia (Duffy et al., 2014).  For long timescale processes, almost every CZO has proposed models of regolith formation, and many are summarized in a special issue (Riebe et al., 2016). At the shorter timescales, standard water or coupled land surface-air models have been tested and new modules developed (Table 3).  To exploit high resolution data such as LiDAR and hyperspectral measurements, modelling efforts explore micro-topographic and vegetation controls on soil moisture (Le et al., 2015; Le and Kumar, 2017) as well as biogeochemical changes in agricultural landscapes (Woo and Kumar, 2017). Researchers have likewise developed a energy-balance snowmelt model that is now being used with remotely sensed data for water supply forecasting (Painter et al., 2016). In other integrative efforts, researchers are modelling how hydraulic conductivity, root water uptake efficiency, and hydraulic redistribution by plants sustain evapotranspiration through dry seasons (Quijano et al., 2012; Quijano et al., 2013; Vrettas and Fung, 2015). Work at the Luquillo CZO has supported interpretations of the controls on bedload grain size and channel dimensions for rivers (Phillips and Jerolmack, 2016).  Researchers at the Calhoun CZO are using distributive models to explore relationships between topographic variations and the landscape's capacity to serve as an atmospheric carbon source or sink (Dialyanis et al., 2015).

Fourth, CZOs act as *baselines to understand and teach about the impact of catastrophic*

*events.* For example, two western CZOs in the U.S. have studied the impacts of wildfire on soil
microbiota (Weber et al., 2014), sediment yields (Pelletier and Orem, 2014), snow accumulation
(Harpold et al., 2014), and water quality (Murphy et al., 2012; Reale et al., 2015). Likewise,
effects of the 2013 Colorado Front Range storm (Gochis et al., 2015) on debris flows (Anderson
et al., 2015), soil moisture (Ebel et al., 2015), cosmogenic radionuclides (Foster and Anderson,
2016), and concentration-discharge behavior (Rue et al., 2017) were studied at the Boulder Creek
CZO.  A flash flood within Boulder Creek CZO in 2016 instigated analysis of Horton overland
flow in these landscapes (Klein et al., 2017). CZOs that experience catastrophic events use the
baseline data captured before the event to place the impact into perspective. An additional
attribute is that such natural disasters engender public interest in research:  research on the 2013
Colorado Front Range storm from the Boulder Creek CZO and wildfire research from three
CZOs has been featured in radio, press, and public forums. The important role of observatories in
recording catastrophic events was reinforced by Hurricanes Irma and Maria, which brought
winds up to 250 km h$^{-1}$ and enormous rainfall to the island of Puerto Rico in September 2017.
The Luquillo CZO quantified winds, rains, and stormflows and will document Maria's impacts to
forest canopies, accelerated tree throw, and mass hillslope movements for many years to come.

Fifth, CZOs act as *the organizing design for systematic campaigns to investigate process-*

*based mechanisms* across different types of CZ. One example of this is the initiative in which
every CZO in the U.S. pursued geophysical measurements. Many papers have been published
exemplifying this approach to map out the subsurface architecture (Befus et al., 2011; Holbrook
et al., 2014; Orlando et al., 2015; Olyphant et al., 2016). Now, geophysicists travel among CZOs
to image the subsurface with a battery of instruments to image the below-ground landscape (St.
Clair et al. 2015). In another example, after the Boulder Creek CZO began emphasizing slope
aspect as a useful natural experiment to examine controls on CZ architecture and function in
2009, similar analyses at other CZOs led to highlighted linkages among aspect, water, biota,
regolith structure, and episodic events (West et al., 2014; Ebel et al., 2015; Pelletier et al., 2017).
Finally, a deep drilling project ("drill the ridge") was proposed and then pursued at many CZOs,
and these data in turn led to a special issue describing regolith formation (Riebe et al., 2016).
Successful campaigns have also been mounted to investigate mountain snow and water balance
(Harpold et al., 2014).

Sixth, CZOs act as *catalysts for the development of new techniques and instrumentation*

which can then be tested globally. For example, at the Eel River CZO, a unique vadose zone
monitoring system (VMS) has been installed consisting of holes drilled into a hill at 55º relative
to the horizontal to monitor for temperature, pressure, and electrical conductivity.  The VMS
probes the generally inaccessible deep vadose zone to test reactive transport models
incorporating gas and water chemistry (Druhan et al., 2017).  At another CZO, experiments
designed to improve management practices for erosion have elucidated controls on the
concentration of carbon in eroded sediment and original soil (Papanicolaou et al., 2015). One
CZO is exploring weathering reactions through the use of neutron scattering (NS) to analyze
pores as small as nanometers in rocks (Navarre-Sitchler et al., 2013). Water-balance
instrumentation using robust wireless-sensor networks, developed at the Southern Sierra CZO
(Kerkez et al., 2012), has been extended to the river-basin scale (Zhang et al., 2017), and is being
deployed at other locations across the U.S. An approach developed to scale annual
evapotranspiration measured at flux towers across the broader forested landscape of the Sierra
Nevada (Goulden et al., 2012) is also being applied to flux-tower sites and forested areas across
the western U.S.

Seventh, CZOs serve as *hubs for informing regional resource-management decisions, and*
*for educating the public about societally relevant problems*. For example, measurements of
evapotranspiration made at one CZO and scaled across the Sierras provide a basis for estimating
sustainable forest densities today and into the future when the climate will be warmer and drier
(Goulden and Bales, 2014). Research on water resources is routinely communicated to water
managers in California and the intermountain west by the Southern Sierra CZO through
briefings, workshops and data products. Results from Catalina-Jemez CZO studies of wildfire
impacts on watershed-scale sediment transport are also being considered in the development of
forest management strategies in two states. Research on snowpack and water resources by the
Boulder Creek CZO has similarly been communicated in a series of workshops for water
managers in Colorado, Utah and Wyoming in 2015.  In other parts of the country, the Eel River
CZO is documenting controls on the spread of cyanobacteria in the Eel River, and information is
disseminated in biannual gatherings of students, agency members, native Americans, and
practitioners. IMLCZO is developing a series of courses for crop advisors in the US agricultural
Midwest. Finally, CZO investigators routinely write op-eds and produce video for distribution to
media audiences and use in pre-college classrooms. For example, the Southern Sierra CZO is a
contributor to the *Sustainable California* web TV channel that was launched with other
collaborators.  CZOs and the national CZO office are active in social media.

Eighth, CZOs act as *incubators that grow innovative teaching and mentor junior*
*scientists* who readily work across multiple disciplines. As shown in Table 1, 39 post-doctoral
scholars worked at CZOs in 2015 along with 106 undergraduate and 186 graduate students. As
more and more institutions in the United States have advertised positions that mention critical
zone science, these CZO students have moved easily into university department faculties where
they are changing the research and education environment. Likewise, the recently completed
InTeGrate project, *Introduction to Critical Zone Science*, is a one-semester undergraduate
curriculum with lecture slides, online resources, and data drawn from the CZOs. This innovative
new course uses the CZ as a unifying approach to teach complex Earth and environmental
sciences (White et al., 2017). Many other teaching and training workshops have also been
presented by the CZOs. For example, a Modeling Institute was presented in 2016 on the *Dhara*
model (Le and Kumar, 2017, Woo and Kumar, 2017) and a training workshop was presented on
the Role of Runoff and Erosion on Soil Carbon Stocks: From Soilscapes to Landscapes in
collaboration with CUAHSI.

Ninth, CZOs act as *impetus for discoveries and emergent hypotheses* that can only result
from systematic and multi-disciplinary observations across multiple CZ environments.  Some of
these hypotheses are disciplinary while others cross disciplines.  A full elucidation of hypotheses
is beyond the scope of this paper and only a subset are shown in Table 4. Many have been
published in collaborative papers (Rempe and Dietrich, 2014; Riebe et al., 2016; Li et al., 2017;
Pelletier et al., 2017; Yan et al., 2017; Brantley et al., 2017, in press). Here we summarize three
multi-disciplinary discoveries that have large implications for the prediction of flowpaths
relevant to the largest supply of accessible and drinkable water available to humans – water
contained in rock and regolith (Fetter, 2001; Banks et al., 2009). These discoveries have been
made both by non-CZO scientists and scientists within a CZO. First, one geophysics group
outside of a CZO discovered a distinct geometry at depth that is consistent with the influence of
regional tectonic stress fields on patterns of fractures and weathering under hillslopes (St. Clair
et al., 2015). The theoretical underpinning proposed for this so-called "bowtie-shaped" geometry
has important implications for predicting flowpaths of water in regolith *a priori*. CZOs also
discovered significant water storage that is seasonally available in the vadose zone of weathered
bedrock (Bales et al., 2011; Salve et al., 2012). This "*rock moisture*", missing from land surface
models, has significant implications for predicting climate. Finally, CZO workers have identified
depth intervals in the subsurface in some sites that document mineralogical reactions and that
roughly mimic the land surface topography albeit with lower relief (Brantley et al., 2013). Such
reaction fronts inform researchers about sub-surface flow paths (Brantley et al., 2017).  All of
these ideas are being tested at other settings around the world.

## 464    6  CZO measurements and models

As mentioned above, common measurements are being made (Table 2) and models are
being used across sites (Table 3). The measurements target the "SWEGS" fluxes – solute, water,
energy, gas, and sediments – as they move through the CZ, as well as such features as the form
and age of the landscape and ecosystems (Fig. 4). Some of the observations are more extensive
than others: for example, hydrometeorology, soil moisture dynamics, and measurements of
concentration and discharge in streams are the focus of on-going efforts at every CZO.

471         The CZOs' datasets are maintained publicly available
(http://criticalzone.org/national/data/) and are intended to serve the research community beyond
those involved in each CZO.  The types of data commonly include sensor and sampler
measurements showing the temporal response of different locations in the CZ to meteoric events,
spatially-resolved geophysical and geochemical measurements of CZ structure, and LiDAR
measurements of vegetation and bare earth topography, among others. The CZOs are
coordinating to ensure that measurements are comparable across sites (i.e., the "common
measurements" effort). Likewise, efforts are ongoing so that the posted datasets can be used
easily by others to make cross-site comparisons and conduct cross-site studies.
The duration of time for individual data sets varies across the network.  Generally, the
time-series datasets (sensor and sampler arrays, eddy covariance, hydrometeorology, vadose
zone and saturated zone aqueous chemistry, etc.) have durations that are roughly equivalent to
the age of the CZO sites, determined by the initiation of NSF funding. One caveat is that CZOs
have often added new study locations that were not among the original set, affecting the time
interval of data that is available at each location. In other cases, measurement series may have
been terminated as new measurements were brought on line.
Three sites (SSCZO, BCCZO and SSHCZO) have been in operation since 2007, and so
their longer-term observational datasets extend roughly over that duration. Three other sites
(CJCZO, LQCZO and CRCZO) that began operating two years later have measurements dating
to 2009. Four newer sites (IMLCZO, CHCZO, RCCZO and ERCZO) have datasets dating to
2013. One observatory (CRCZO) ceased functioning as a CZO in 2014. Therefore, at present,
continuous time series datasets range in duration from ca. 4 to 10 years. In addition, however,
several of the CZOs are located in sites that provide longer datasets through previous
measurement programs (for example, the extremely long datasets available at the Reynolds
Creek CZO). The nature of dataset duration is thus somewhat complex, and varies depending
upon data type and site, but the generalized intent is to enable assessment of inter-annual
variation over decades. The datasets are starting to drive extrapolations from the individual study
sites to regional and continental scales. The duration of datasets also depends upon the residence
times and mixing times of the various measured entities (Fig. 4).To integrate the measurements
at different sites and to extrapolate forward and backward in time requires process-based
modeling. As the common observational data accumulate, CZOs have been both developing new
models and pursuing data comparisons with established models (Table 3).  Currently, the initial
CZ modeling efforts may be characterized into four groups as discussed below (Table 3).
The first includes the modification and coupling of existing codes to link various CZ
processes (e.g., land-atmosphere exchange, saturated-unsaturated zone hydrology,
biogeochemistry, ecology, etc.) that are typically segregated in distinct models, but whose
coupling is being revealed through CZ science measurements. The second includes identifying
and filling critical gaps or knowledge of new processes such as hyporheic exchange, weathering,
etc. The third includes development of a new generation of models that takes advantage of
emerging streams of high-resolution data such as airborne- and UAV (unmanned aerial vehicle)-
based LiDAR and hyperspectral data. The fourth includes coupling between fast and slow
processes across many time scales. Slow processes provide the template for the fast response
variable, while the accumulative effect of the latter results in the evolution of the former. Both
mathematical frameworks and data to support such modeling are still in their infancy.
**7  Emergent network-level concepts**

A central challenge of CZ science is the need to generalize from the place-based studies at observatories to principles-based understanding across the network or across the globe. One way to do this (perhaps the only way) is with models. Dialogue is ongoing as to whether the critical zone community will be best served through a single modeling framework or a library of existing models that allows more targeted exploration. Place-based studies can demand very specific investigations that are highly tuned to the biogeomorphic setting of a specific location, but that provide little deeper understanding. In contrast, a model that is broadly applicable may simplify the representation of a given site so much that the model results in reduced accuracy of prediction. Therefore, both the advancement of critical zone science and critical zone modeling will likely progress in an intertwined manner.

One way to further the evolution from place-based to principles-based understanding is to drive development of fundamental understanding at a network level, rather than the level of a single observatory. In fact, since initiation of the CZO effort in 2007, three general, overarching concepts have emerged at the network level. Each of these describes deeper process- and principles-based understanding as summarized below.

First, we have observed that differences in natural and anthropogenic inputs at Earth's surface translate into differences in water, regolith structure, minerals, and biotic activity at depth, and we are starting to detect how these deep properties also impact the biota, climate, and CZ services (e.g. Richter and Billings, 2015; Sullivan et al., 2016; Richardson and Kumar, 2017; Chorover et al., 2011).

Second, we have observed how the deep surface of the Critical Zone varies across landscapes. Under hills, imaging has revealed locally consistent patterns of subsurface critical zone structure that can relate depth, fracture density, porosity, and weathering (e.g. Befus et al., 2011; Brantley et al., 2013; Orlando et al., 2015; St. Clair et al., 2015).

Third, we now have mechanistic models that provide quantitative predictions of the spatial structure of the deep surface relative to the ground surface topography (e.g. Lebedeva and Brantley, 2013; Rempe and Dietrich, 2014; Rasmussen et al., 2015; Riebe et al., 2016). These three broad generalizations have been informed by up to ten years of work at multiple CZOs as well as work by the greater critical zone science community (Banwart et al., 2011).

In addition to the emergence of these network-level science concepts, an important link has emerged between the CZ and the concept of "ecosystem services". This concept emphasizes how biodiversity, ecological processes, and spatial patterns in the near-surface environment provide services to society (MEA, 2005). As discussed by CZO network scientists (Field et al., 2015; 2016), CZ science demonstrates the contribution of the deeper CZ to ecosystem "provisioning" and elucidates the longer time scales of CZ evolution, leading to the idea of "critical zone services". Through this lens, services such as water quality regulation, soil development, and carbon stabilization are seen as tightly dependent on CZ function, evolution

and architecture. The valuation of CZ services offers an approach for assessment of human impact that takes into consideration both the short- and long-time scale processes ((Richardson and Kumar 2017). Indeed, the CZ is the ideal context for integrating deep subsurface and long time-scale perspectives from geosciences into the otherwise bio-centric conceptualization of ecosystem services. Doing so remains an emphasis of CZO network activities.

## 8 Strengths and weaknesses of the current CZO network

The current CZO network as constituted in the U.S. and abroad has many strengths. Students are trained to cross disciplines within their work, and they graduate with convergent expertise in the new field of CZ science. CZ science is harmonizing vocabulary and conceptual understanding across disciplines, and is setting a research agenda and an integrated approach. Postdoctoral scholars learn from observatory personnel that derive from many disciplines. Such scientists now communicate as effectively about sapflow in trees as about seasonal variations in groundwater flow. Collaborations are constantly developing and allowing scientists to see problems with different perspectives. Ideas are growing across the network about regolith formation (Riebe et al., 2016), snow hydrology (Harpold et al., 2014; Tennant et al., 2017), microbial diversity (Fierer et al., 2003), trees (Brantley et al., 2017, in press), and many other topics. We have produced enormous datasets (http://criticalzone.org/national/data/datasets/). We no longer treat parts of the CZ as isolated components or black boxes: instead, we incorporate more specificity and understanding when we describe the integrated system. We are finding innovative ways to communicate the CZ concept to the public. We have stimulated and promoted development of CZOs worldwide.

Although there have been many successes, we also observe weaknesses. Since each observatory is individually funded based on the merits of its targeted science, there is competition for allocation of resources to address common measurements versus site-specific activities. This results in a less-than-optimal identification of emergent network-scale outcomes. Of course, individual site-specific outcomes can have implications and impacts that are just as important as network-scale outcomes. Thus we need to find mechanisms to foster all such approaches while acknowledging limitations in resources. Further, given how new CZ science is, insights are only just beginning to emerge that cut across multiple disciplines. We still have occasional difficulty communicating these ideas in a simple fashion. One specific example of a need for cross-disciplinary ideas and communication arises from the fact that the CZO network in the U.S. never emphasized social science. Thus, hypotheses have yet to emerge that target social science aspects of the CZ. Another challenge, and perhaps our biggest, is maintaining the integrity of an inter-disciplinary suite of measurements in a common database and managing site data (Fig. 4) in ways that invite other researchers to find and use the datasets (Hinckley et al., 2016). The need for better data management is especially important given the many new data-driven approaches that are arising within environmental science (Bui, 2016).

These considerations in the context of the overview in this paper lead to a basic question:
*how might we design the best mechanism to advance CZ science?* We point to four specific
challenges, posed here as questions, that loom large in designing the future network. First, what
is the best approach to developing broadly-applicable principles from observatory-based
investigations? Second, how do we link appropriately with other programs in the U.S. and
worldwide to develop a set of representative sites across the large number of possible
environmental gradients to advance a broad understanding of CZ science? Third, how should we
balance the roles of CZOs in developing long-term observational records versus shifting
measurement strategies to advance and test new hypotheses?  Fourth, what funding and
management models would enable increased involvement of CZ scientists who are not yet part of
core CZO teams? These four issues are addressed in the next section where we propose a new
model for the future of the network.

## 9   The future network

Mechanisms that have been successful in stimulating deeper understanding of the
environment were described above. These strategies can be summarized as i) small investigator
projects targeting parts of the CZ, ii) campaign-style multi-investigator projects targeting
multiple sites, iii) center-based efforts, and iv) observatories. Looking into the future, all are
needed.
For example, individual grants (example (i) from above) can test sharply focused
hypotheses that may lead to important discoveries about individual entities or processes.  This
kind of research, typically supported by a core grant from within a specific discipline (e.g.,
hydrology) sustains both the discipline and advances CZ science.  The last decades of research
has clearly shown that some advances come from single investigator research.
Mechanism (ii), campaign-style research, has the advantage of exploring CZ questions
over a range of conditions (Larsen et al., 2015).  Such campaigns often focus on material
properties or process dynamics. Campaign-style research can focus many different one-time
measurements, or measurements across a larger spatial scale, than CZOs can routinely
accomplish. Campaigns typically incorporate small teams of researchers.
Establishment of centers (approach iii) is another important means to guide and test field
data collection and to probe for deeper understanding by fostering communication and
collaboration among many researchers. However, the CZO approach (iv) is the only approach
that forces diverse researchers to tackle fundamental questions at a single location while also
performing the nine important roles described above. In particular, only observatories provide
the long-term data and the diverse co-located observations from all disciplines that we need to
understand the CZ. By working together with centers, only place-based observatories can knit
together disparate views by acting as gathering points for scientists from all disciplines with all
their skills, instruments, and models.
However, because the CZ is highly heterogeneous, the network must be designed to
promote the emergence of informative ideas that supersedes this heterogeneity. In other words,
CZOs must collaborative to engender  network-level insights. Given this need, one approach
may be for the community to identify a broad common-question for the future and then to design
the future network to target this overall question.  One proposed example for the next decade of
CZO research is the following question of central importance:
*How can we increase our understanding of surface and subsurface*
*landscapes and fluxes as we face climate, land use, and other*
*anthropogenic changes in the future?*
With such a question, the entire CZO network could test sub-questions and sub–hypotheses
together, but with experiments at different sites with different characteristics.
Even if the CZOs target this question together, some adjustments to the current topology
of the CZO network should be evaluated and updated so as to promote the emergence of
network-level ideas. Indeed, many other topologies can be imagined (Fig.  5). Many scientists
have similarly considered aspects of what is needed for environmental networks (Leopold,
1962). For example, one topology might be to choose new observatories to fill in gaps among the
current nine CZOs, as shown schematically in the diagram. Another model might be to continue
the current nine CZOs as the future network in order to sustain both their unique observational
records and the theoretical advances these advances enable. Another model might be to choose
nine (or some other optimized number) completely new CZOs, i.e., treating the country as a
blank slate. Another model might be to complete a careful analysis of the current CZOs in the
context of Long Term Ecological Research (LTER) sites and Long Term Agricultural Research
(LTAR) sites and then to extend the network appropriately. NEON should also be part of this
leveraging, as it becomes operational nationwide. A fifth model might be to establish various
"hub" locations and then choose smaller sites along environmental gradients extending out from
the hub. Finally, many CZOs might be funded for research along with smaller satellite sites that
extend from the central CZO.
In thinking about the future network topology, we emphasize the need to find solutions to
the four problems stated as questions at the end of the last section: 1) the need to advance
principles-based understanding from observatories; 2) the need to coordinate with other US
programs and CZOs world-wide to sample a wide range of CZ conditions; 3) the need for
balance between measurements for hypothesis-testing and common, core measurements as
network infrastructure; and 4) the desire to incorporate an even broader community of
researchers into the CZO program.

The best topology to address these issues is a design like the hub-and-spoke model but with multiple hubs and a high degree of scientific coordination with the other networks noted above. In addition, instead of spokes, i.e., lines of satellite sites that extend geographically out from the hub, we prefer to call these "campaigns", noting that in some cases these satellites might indeed be spokes, but in other cases they might be located in vastly disparate locations. This model would answer the need for long-term measurements (at carefully chosen hubs), the need for short-term targeted measurements at specific locations both within the U.S.A. and abroad (carefully chosen campaign sites), and the need for new mechanisms to engage more investigators (funding to bring in scientists from outside the hub network). This long-term hub and short-term campaign emphasis has been promoted in the literature by researchers both inside and outside the current CZO funding framework (Banwart et al., 2012; Larsen et al., 2015). Fundamentally, the argument for the hub and campaign approach is that the two methods benefit from each other. Specifically, the hub provides the unique opportunity to dig deep into understanding mechanisms and process dynamics, whereas the campaign approach provides an opportunity to test the generality of specific findings, ideas, or theories across some relevant gradient, e.g., in climate, land use, or tectonic activity while bringing in outside researchers.

Specifically, we propose a "hubs-and-campaign" network that would consist of several (or all) of the CZOs as hubs that would provide the infrastructure for common measurements – and would be stimuli for ephemeral campaigns funded for shorter periods with more constrained purposes that incorporate non-CZO personnel. The hubs would perform all nine of the CZO roles listed above and would receive stable funding. In contrast, the campaigns would be funded in efforts for shorter time periods to test specific hypotheses or ideas. In this way, the network would be able to change appropriately with time and cover more environments, would provide both infrastructure and hypothesis testing, and could be nimble and inviting for more groups to participate.

It makes sense for the hubs to be located in settings of broad interest from a scientific and societal point of view. For example, an urban CZO could be considered. Alternately (or in addition), the hubs could be located to test specific hypotheses about critical zone structure and controls across gradients of attributes such as climate or tectonic activity or disturbance. Hubs, chosen for their strategic and scientific importance, would presumably also be chosen in recognition of the needs of human resources for education and outreach, and of the need for both applied and curiosity-driven science. The potential use of hubs as attractors for students and scientists and platforms for increasing diversity in the Earth and environmental sciences would need to be stressed.

## 10 Conclusions

We now recognize the critical zone as an entity composed of co-evolving systems that create the structured dynamic skin of the Earth. We are seeing the first maps of this structure as they emerge and we are discovering how the structure influences water resources and hydrologic

processes, vegetation, ecosystems, erosion, biogeochemical processes, and even regional climate.
Surface and deep processes are connected. A first set of testable models has emerged and now
points to specific measurement programs. But this is only the beginning. While progress has
been made, the central questions remain: what controls the critical zone properties and processes,
how does the critical zone respond to climate and land use change, and how can we use our
advancing understanding to benefit societal needs? These fundamental questions will require a
sustained research commitment. The critical zone is a frontier area of science where only the first
observations have been obtained. New methods, instrumentation, and theory are needed to
continue to grow convergent understanding.
Future research in critical zone science will be best advanced through a combination of
distributed long-term observatories strongly coupled with focused, campaign-style
investigations. These campaigns would target new sites that might radiate from the central hub
observatory to test specific hypotheses and theories across controlling gradients. The
observatories would focus on the necessary long-term monitoring to reveal mechanisms and
dynamics. The field campaigns would collect data over shorter periods.
The decision by the US National Science Foundation to support a network of Critical
Zone Observatories since 2007 has laid the foundation for a new discipline of critical zone
science that has driven the convergence of individual scientific disciplines. Former graduate
students supported at the CZOs are now taking up faculty posts and rapidly introducing new
courses that span the many disciplines needed to reveal critical zone workings. The next
generation is in the making. Findings from the CZOs are being absorbed by agencies and put to
practice. The power of the critical zone concept has spread across the globe and stimulated the
building of numerous critical zone observatories. We are seeing just the beginning and it is time
for the next chapter.
**Acknowledgement**s. We acknowledge the help of S. Sharkey (funded by the CZO Science Across
Virtual Institutes Project), T. Bernier, and D. Lambert, and funding from NSF grants EAR 13-
31726 to S.L. Brantley, EAR 13-31906 to P. Kumar, EAR 13-31872 to K. Lohse, EAR 13-31846
to Richter, D., EAR 13-31408 to J. Chorover, EAR13-31828 to S.P. Anderson, EAR 14-45246 to
T. White, EAR 13-31841 to W.H. McDowell, and EAR13-31940 to W.E. Dietrich. The manuscript
benefitted from reviews by J. Tunnicliffe and an anonymous reviewer, and comments from K.
Bishop, P. Schroeder, and E. Bui and editorial handling by J.M. Turowski.

Table 1. Metrics enumerating the U.S. CZO experiment

| | |
|---|---|
| CZOs in the United States | 9 |
| CZOs worldwide | 45* |
| Countries with interdisciplinary field observatories hosting CZ science** | 25 |
| Papers citing critical zone in keyword WOS as of 2017[#] | 926 |
| Papers listing critical zone in title in WOS as of 2017 | 242 |
| Post-doctoral students educated at CZOs in 2015 | 39 |
| Graduate students educated at CZOs in 2015 | 186 |
| Undergraduate students educated at CZOs in 2015 | 106 |

*Includes Germany (TERENO), France (RBV/CRITEX), UK, and China

**Giardano and Houser (2015)

[#]Papers returned through searching Web of Science (WOS) as of 05/28/2017 that include "critical zone" in title or key word or abstract etc. (not including abstracts for meetings)

Table 2. Common measurements made at the CZO network in the U.S.A.

| Critical Zone Observatory – Measurement Type | Boulder Creek | Cal-houn | Catalina-Jemez | Eel River | Intensively Managed Landscapes | Luquillo | Reynolds Creek | Susque Shale Hills | S. Sierra |
|---|---|---|---|---|---|---|---|---|---|
| **Land-Atmosphere Exchange** | | | | | | | | | |
| LIDAR | X | X | X | X | X | X | Y,Z | X | X,Y |
| Eddy flux | Y | Z | X | | X,Y,Z | | Y | X | X,Y,Z |
| Wind speed and direction | X | Z | X | X | X,Y | Y | Y | X | X |
| Precipitation and throughfall | X | X,Z | X | X | X,Y | Y | Y | X | X,Y |
| Wet deposition and bulk deposition | X | Z | X | | Y | Y | Y | x | X,Y |
| Snowpack distribution and duration | X | | X | | | | | | X |
| **Vegetation and Microbiota** | | | | | | | | | |
| Structure and function above and below biomass | X | X | X | X | X | Y | Y,Z | X | X,Y |
| Microbial composition | X | X | X | X | X | Y | X | x | X |
| ET-species composition and structure relationships | Y | Z | X | Y | Y | | Y,Z | X | X |
| **Soil (Vadose Zone)** | | | | | | | | | |
| Solid - elemental composition and mineralogy | X | X | X | X | X | X | Z | X | X,Y |
| Solid - texture and physical characterization | X | X | X | X | X | X | Z | X | X,Y |
| Solid – organic matter content | X | X | X | X | X | X | Z | X | X |
| Solid – radiogenic isotope composition | X | X | X | | X | X | | X | X,Z |
| Fluid – soil moisture (sensors) | X | X | X | X | X | X | Y,Z | X | X |
| Fluid – soil temperature (sensors) | X | X | X | X | X | X | Y,Z | X | X |
| Fluid – soil solution chemistry (samplers) | X | Z | X | | X | X | | X | X,Y |
| Fluid – soil gas chemistry (samplers/sensors) | | X | X | X | | X,Y | Z | X | X |
| **Saprolite and Bedrock (Saturated Zone)** | | | | | | | | | |
| Solid – petrology and mineralogy | X | X | X | X | Y | X | Z | X | X,Z |
| Solid – elemental composition and OM content | X | X | X | X | X | X | Z | X | X,Z |
| Solid – texture, physical/architectural constraints | X | X | X | X | X | X | Z | X | X |
| Fluid – potentiometric head, temperature (sensors) | X | X | X | X | X | X | Y | X | X |
| Fluid – groundwater chemistry (samplers/sensors) | X | X | X | X | X | X | Y | X | X |
| Fluid – saprolite/weathered bedrock gas chemistry | | X | Z | X | | | | x | |
| Geophysical surveys – depth to bedrock | X | X | X | | X | X | | X | X |
| **Surface Water** | | | | | | | | | |
| Instantaneous discharge | X | X | X | X | X,Y | X,Y,Z | Y | X | X,Y |
| Stable isotopes of water | X | X | X | | X | X | | X | X |
| Stream water chemistry (samplers/sensors) | X | X | X | X | X | X,Y,Z | Z | X | X,Y |
| Sediments (samplers/sensors) | X | X | X | X | X | X,Y,Z | Y,Z | X | Y,X |
| Extent of wetted channel | | | | X | | | | | Y,X |
| Aquatic biota (invertebrates, fish, etc.) | | | | X | | Y | | | Y |
| **Age or rate constraints** | | | | | | | | | |
| Cosmogenic radionuclides | X | X | X | | | X | | x | X |
| C ages | X | X | X | | | X | | x | |
| Optical Stimulated Luminescence ages | X | | | | | | | | |

x: instrumentation in place or sampling is occurring, owned and operated by the CZO; y indicates instrumentation is currently in
place, owned and operated by a partner of the CZO; z indicates that it is planned to be installed or implemented in the future by
the CZO

Table 3: Models used by the U.S. CZOs (observatory abbreviations in Fig. 2)

| Model name | Systems modeled | Possible X-CZO appli-cation? | CZOs using model |
|---|---|---|---|
| PIHM | Hydrology | x | CR, SH, CL |
| Flux-PIHM | Hydrology, land/atmosphere | x | SH |
| tRIBS | Hydrology | | LQ, CL |
| hsB-SM | Hydrology | | CJ |
| VS2D | Unsaturated hydrology | | BC |
| Dhara | Near surface critical zone | x | IML, SH |
| Optimal sensing | Soil moisture | | CL |
| Hydropedo Toolbox | Soil moisture | | SH |
| OTIS | Streambed hydrologic exchange | | SH |
| Alpine glaciers 1&2d | Ice motion | | BC |
| Fluid exchange | Estuary fluid flux | | CR |
| PHREEQC | Aqueous geochemistry | | LQ |
| WITCH | Weathering | x | SH, CL |
| ROMS | Ocean | | EL |
| WRF | Weather forecasting | | EL, RC |
| ISNOBAL | Snowcover mass | | RC, SS |
| SHAW | Heat and Water fluxes | | RC |
| CHILD | Erosion, sediment transport, surface evolution | x | BC, CL |
| SOrCERO | Erosion and deposition | | CL |
| Digital glacier bed | Elevation of glacier bed | | BC |
| Gully Erosion Profiler | Channel profile evolution | | BC |
| Hillslope Trajectory | Erosion | | BC |
| Range and Basin | Mountain evolution | | BC |
| Landlab | General 2d models | | BC |
| FEMDOC-2D | Hillslope DOC transport | | CR |
| IDOCM_1D | Heat and DOC in soils | | CR |
| CENTURY | Soil carbon | | LQ |
| CN reforest dynamics | Tree/soil C and N | | CL |
| BIOME BGC | Carbon | | SH, RC |
| Plant-soil feedback | Plant-soil, soil production | | CL |
| Root deformation | Soil deformation from roots | | BC |
| GASH | ET and throughfall | | LQ |
| NPZD | Ecosystem | | EL |
| PIHMSed | Water/sediment transport, uplift, weathering | x | SH |
| PIHM-DOC | Hydrology, dissolved organic carbon | | CR, SH |
| TIMS | Hydrology+microbio/geochem/geomorph/ecology | x | CJ |
| RHESSys | Hydrology, ecology | x | SS |
| AWESOM | Atmosphere/watershed/ecology/stream/ocean | x | EL |
| tRIBS-ECO | Hydrology, erosion, soil C | | CL |

| | | | |
|---|---|---|---|
| **WEPP/CENTURY-WEPP/WEPP-Rill1D(3ST1D)** | Soil erosion, biogeochemistry | x | IML |
| **OpenFOAM/BioChemFOAM** | Riverine transport | x | IML |
| CRUNCH | Reactive transport | x | SH, IML, EL |
| Delft3D | Surface/subsurface transport | x | EL |
| Nays2D | Flood | x | IML |



Table 4. A few emergent hypotheses from the CZO network
1) CZ architecture controls hydrologic and geochemical processes that drive concentration-
discharge relationships in rivers.
2) The depth to fresh bedrock across uplands landscapes may be predictable from models that
account for regional stress fields, advancing chemical reaction fronts, drainage of the fresh
bedrock, and/or fracturing from freeze-thaw.
3) Aspect differences can be used to reveal the mechanisms and effects of climate on the CZ.
4) The deep microbial community is linked to overlying vegetation: microbial community is
distinctly different under agriculture fields, brush, grassland, perennial forest and deciduous
forest.
5) The deep microbial community is linked to lithology: the microbial community is distinctly
different on granite, basalt, shale, and sandstone.
6) The deep architecture of the Critical Zone controls water availability to plants and microbial
communities, which in turn influence regional climates.
7) Subsurface reaction fronts may often be used to map flow paths in the subsurface.
8) Human impact in intensively managed landscapes has resulted in a critical transition that has
changed the landscape from primarily a transformation-dominated system characterized by long
residence times of water, carbon, and nutrients, to a transport-dominated system characterized by
fast movement of water, sediment, carbon, and nutrients through the landscape into receiving
water bodies.

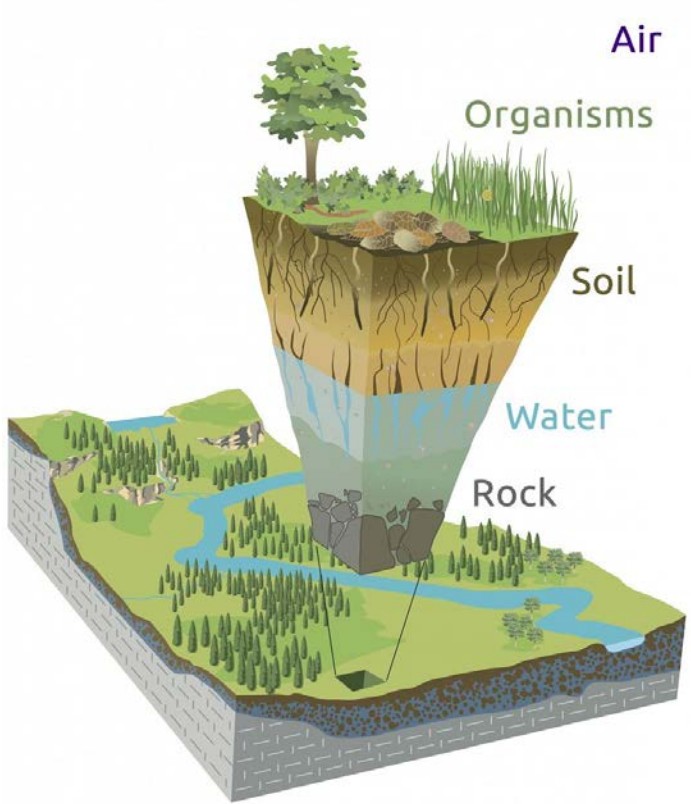

Figure 1. Understanding the critical zone requires harnessing insight from many disciplines on processes and fluxes from the top of the vegetation canopy down into groundwater at all spatial scales across timescales from milliseconds to millennia.   Figure reproduced from Chorover et al. (2007), artwork by R. Kindlimann.

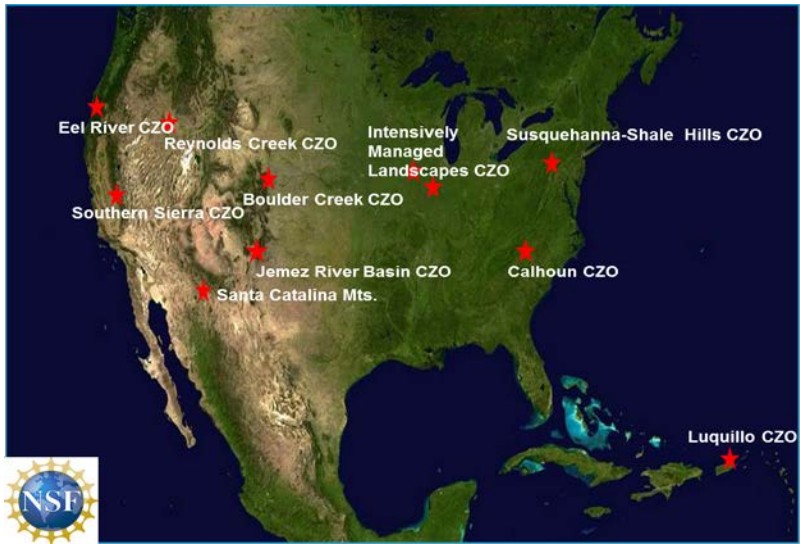

Figure 2. The current network of nine Critical Zone Observatories funded in the United States to investigate all aspects of the critical zone. Abbrevations used in Table 2: Eel River (ER), Southern Sierra (SS), Jemez Catalina (JC), Boulder Creek (BC), Reynolds Creek (RC), Intensively Managed Landscapes (IML), Susquehanna Shale Hills (SH), Calhoun (CL), Luquillo (L).

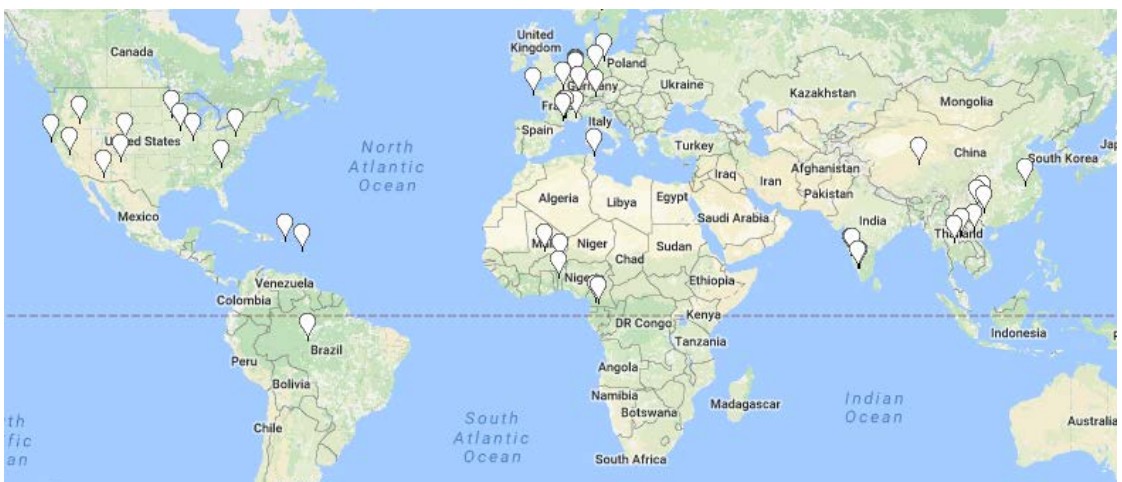

Figure 3. Location map of the 45 CZO locations in the U.S., Germany (TERENO), France (RBV/CRITEX), UK, and China that have been registered on Site Seeker (http://www.czen.org/site_seeker). The sites outside of Europe that are not in China are all operated by the French RBV/CRITEX program.

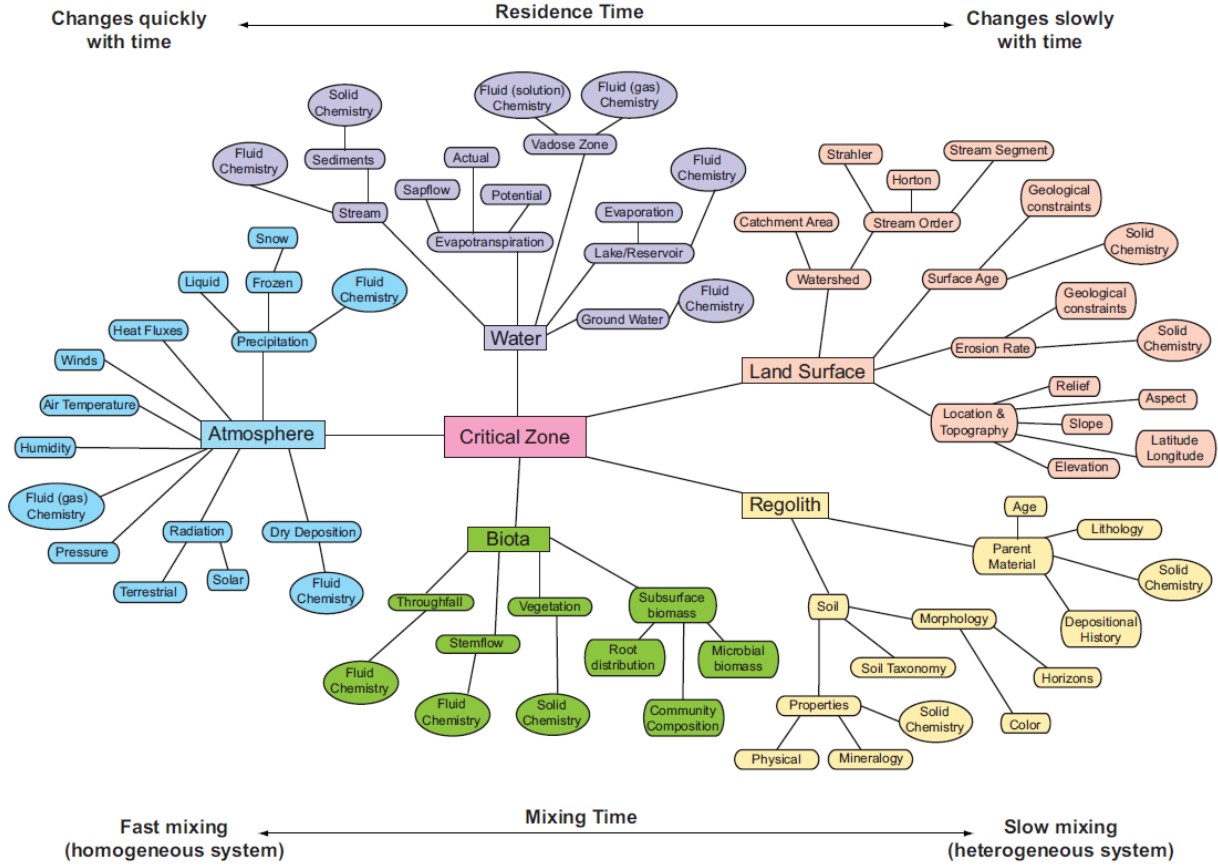



Figure 4. A schematic diagram showing some of the major entities that can be measured as part
of the critical zone. The colors code entries related to the atmosphere (aqua), water (indigo),
land surface features (beige), regolith (yellow), and biota (green). As shown by arrows, the
entities are organized on the diagram from short to long residence times (left to right
respectively), and these correlate with generally fast to slow mixing times respectively.
Reproduced with permission from Niu et al. (2014).


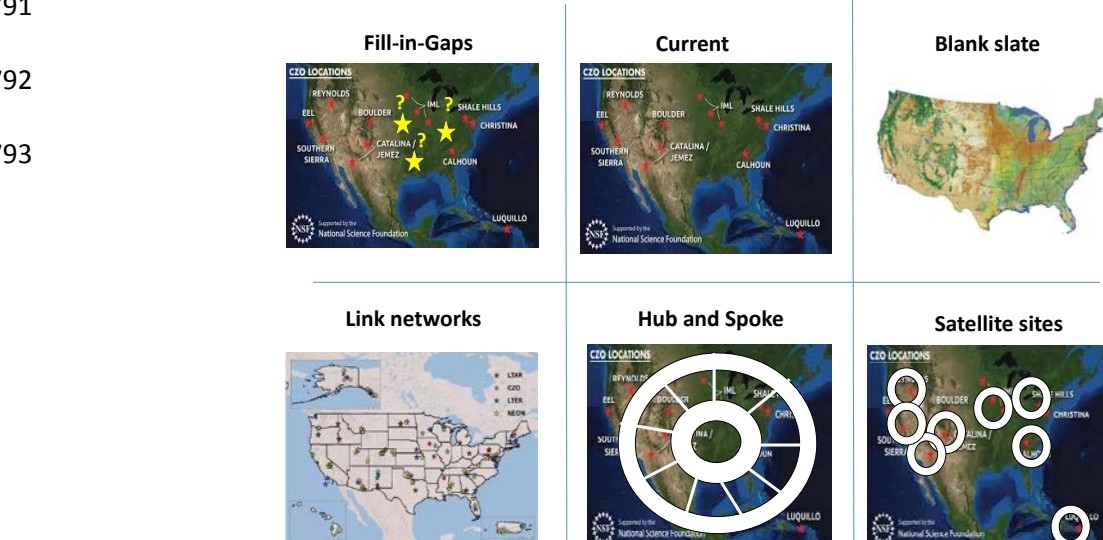

Figure 5. Conformations of a future CZO network discussed in the text. . The most effective topology is likely to be a combination of the observatory framework with smaller campaign-style science as discussed in the text, i.e., a "hubs-and-campaign" strategy.

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
