# Peer review of "Designing a network of critical zone observatories"

_Earth Surface Dynamics, 2017_

## Short Comment (SC1) · 3 Jul 2017

At a recent Gordon Conference that gathered over a hundred members of the catchment science community, it was striking to see how the important the work done at CZO's has become to catchment science and the issues addressed about the sustainability of water resources and aquatic ecosystems. This importance is even more striking when thinking how many of the contributions from outside the US were from study areas that could fit in well with some further expansion of the CZO concept. Thus this paper, with a look to the future of networking CZO's is a very timely contribution.

[Figure]

2017.

**ESurfD**

Interactive
comment

---

## Short Comment (SC2) · 13 Jul 2017

Included in the observatory network should be urban sites (>50,000 people). Furthermore, urban sites that have been continuously influenced by humans on millennial time scales (>5000 years). There are very few places that satisfy this later criterion but perhaps it is worth considering.

---

## Short Comment (SC3) · 14 Jul 2017

The CZ conceptual framework has advanced our understanding of Earth surface processes with more rigour than traditional pedogenesis and landscape ecology studies and more detail than current land surface exchange models can accommodate. Thanks to CZ science we have a better understanding of weathering and landscape evolution, and interactions between tectonics and Earth surface processes. Perhaps the most valuable contribution the NSF-funded CZOs have made is the training of students who are able to think about Earth surface systems and their components beyond a single disciplinary viewpoint—hopefully they can all find satisfying jobs. CZ science

is transciplinary and transformative. To build on its success the NSF should emphasize CZ science that: 1) can address current 'wicked' societal problems and help formulate better land development and environmental management policies. That means more studies in intensively managed landscapes, urban landscapes, landscapes exploited for their mineral, oil, and/or gas resources—and more explicit linkage with social sciences. 2) facilitates extrapolation from CZO-based science and put the results in broader regional and continental context. This means working with researchers like me who build predictive spatial models of soil and geochemistry over large regions. Spatially distributed reactive transport models would be the ultimate objective here and would enable the next. 3) aims to predict the change trajectory that Earth systems might take under global warming (as proposed in the paper). This will require working with land surface modellers to refine the scale of their models.

---

## Referee Comment (RC1) · J. Tunnicliffe (Referee) · 16 Jul 2017

I enjoyed reading this summary of the many insights and successes that have come about through the use of a 'Critical Zone' science framing. The paper is well-suited to ESurf readership, and the writing is generally clear, concise and free of grammatical issues. I believe the structure could be improved to help readers who may not be familiar with the CZ science framework; I have provided some general comments that may help with the flow of ideas.

(1) The organization of the first 3-4 sections could be improved: Section 1 provides an overview of the CZ science framework and summarizes the many metrics of success

for CZ science. Section 2 begins as a review of 'investigative approaches' (line 112), but then veers into models of funding research (e.g. single year, single investigator vs. multi-year, multi-researcher campaign vs centralized funding). I was anticipating a summary of the intellectual heritage of inter-disciplinary CZ science, rather than a review of evolving government funding strategies (at this point in the paper). The former is surely more important in focusing and developing the objective of this review, i.e. developing the observatory network. The evolution of funding should come later, building toward the future proposed design of the network. Section 3 provides the intellectual context that I was looking for, and I think this should be brought forward: the idea of place-based and integrative science should be introduced early, particularly as it relates to the evolution of scientific thinking. The emerging framework's deep roots in applied research are brought out here, as well as the important links to social science. In Section 4, the four common elements of CZOs are illuminating (p.8 ln. 273), and these could be given a more prominent place, earlier in the paper.

(2) I don't see that the title fully matches the content of the paper, in its current form. This is an ambitious retrospective review of the Critical Zone framework with its many strengths and opportunities, but the question of 'designing an observatory network' only makes an appearance in the latest stages of the paper (Section 8). This could be remedied with some re-organization of the first 3-4 sections of the paper, reminding the reader of the intent to 'develop a network', with a stronger framing of the text.

(3) CZ science is referred to as an 'interdisciplinary experiment', but there is very little context provided in lines 87-92 and 113. I presume the term 'experiment' is being used colloquially? If so, perhaps it is better described as a venture, endeavour or initiative. At any rate, some clarification of terminology is needed, here. Such terminology is important when working in inter-disciplinary spaces.

(4) The word 'paradigm' is bandied about rather loosely, in my opinion, without adequate consideration of the philosophical criteria for true paradigm change. The Critical Zone approach might be considered a new paradigm (line 79), though methodological

innovations that come about through successful inter-disciplinary work do not necessarily constitute true paradigm change. You should convince the reader that the criteria have been met, in a succinct summary. The three 'emergent paradigms' in Section 6 are tantalizing and novel, but I do not see how they constitute new paradigms, at least, not as they are explained. The list of emergent hypotheses in Table 4 are similarly intriguing, but they are not 'transformative', in the sense that they are pursued using fairly conventional scientific epistemology. Nonetheless, the paper makes an excellent case for the transformative potential of the CZ approach, and will be helpful in stimulating the conversation regarding next steps for this exciting venture.

Some Specific Comments:

Pg 10, ln 355: "..to investigate process-based mechanisms in across various CZ environments". Could say 'process-based changes' or simply 'mechanisms'. Note: 'in [and] across various CZ environments' Figure 1 appears to be a fairly conventional watershed science diagram. Could it further emphasize canopy and biological elements? Figure 3 caption mentions five countries, but pins are placed in a dozen or more. Explain the relationships. Table 2 does not back up your point about long-term measurements. It would be more helpful to see the length of these records, rather than a smattering of similar measurements that may or may not relate to broader hypotheses being tested across CZOs. Pg 9, ln 307 makes reference to the "extremely long" duration of the datasets - this could use some quantification. Table 3 seems to catalog a broad array of models, most of which were not specifically designed for CZ research. It would be more interesting to list emerging numerical models, or amalgamations of existing models, next to the specific CZ questions being pursued. Table 4 is missing any mention of hypotheses related to the social science aspects of the CZ ('human impacts' aside), one of the stated strengths of the approach. It would be good to see how this strand of the research fits in!

---

## Referee Comment (RC2) · Anonymous Referee #2 · 25 Jul 2017

The paper from S.L. Brantley et al. "Designing a network of critical zone observatories to explore the living skin of the terrestrial Earth" is a review of CZ research leading to a suggestion of a design for future studies and research. The structure of the paper is based on eight sections: the first paragraph introduces the concept of the CZ; the second explains the brief history of research funding; the third paragraph explores the CZO networks over the last decades; fourth paragraph enumerates a series of science field involved in the CZO research; fifth paragraph explains the creation of network in the U.S. through research programs; sixth paragraph presents the main roles of the CZO's in today's research; seventh paragraph exposes emergent paradigms; finally eighth paragraph concerns authors thoughts on future design for efficient research on

the CZO. The CZ research is presented in an evolutive approach of broad interest for growing science community. The hindsight of CZ research is within the scope of the journal and should be considered for publication in Earth Surface Dynamics. The article is mostly free of grammatical errors and writing is clear. The paper addresses relevant scientific questions based on numerous bibliographical references and reaches a proposition for a novel concept of scientific interest.

The structure of the paper could however be improved for the reader to better understand the process leading to a design future CZ network. To improve readability, I would suggest better separation between the third and fourth paragraphs. I would suggest the paragraph 3 to explain the first research initiatives and evolution to resulting CZO programs with main research achievements. Paragraph 4 could then explain the limits of those research programs and the potential field to be developed in the future to go beyond the past achievements. The CZO roles presented in paragraph 5 would then be introduced more clearly.

Specific comments: Paragraph 7 mentions briefly the publication of numerous datasets (p.7 l.1) sometimes spanning several decades of measurements. I feel the creation of this repository as well as the website is a very important achievement in the CZO programs and should be highlighted in the text. This repository can also open new opportunities to future research based on those datasets and be a valuable asset for the modeling community and for the communication to the public. In my opinion, the future of the CZO research will also go through this step in order to better understand variability between different locations and relation with anthropogenic or climate change.

P :12, l.24 : Remove the uppercase from "The"

P15, l.6: remove "to be"

Table 2, 3 and 4: Those tables represent valuable achievements of the CZO research but are only briefly described in the captions. They should be a little more explained in the text. Figure 3: The locations of pin points on the map identify CZO in many other

**ESurfD**
countries than just the ones from U.S., Germany, France, UK and China. The caption should be completed or the figure should be modified to locate only CZO from those countries.

**ESurfD**

---

## Editor Comment (EC1) · JM Turowski (Editor) · 25 Jul 2017

Dear authors,

we have now received the two solicited reviews and a number of further short comments. In general, your effort in writing this paper is appreciated and the paper seems to be a welcome contribution to critical zone science. However, both reviewers highlight that the stsructure of the paper is not optimal and could be improved. There are also a number of minor comments to improve clarity and ammend the content.

Please provide a detailed reply to all comments.

[Figure]

With best wishes,

Jens Turowski
* * *

---

## Author Comment (AC1) · 8 Sep 2017

Authors' response: Esurf-2017-36

We appreciate the comments on our paper reviewing the arc of CZ research and proposing a strategy for the future design of an observatory network. We found the reviewers' comments thoughtful and helpful as discussed below. We also appreciated the comments from outside the U.S.A., specifically reviewers from Sweden (K. Bishop) and Australia (E. Bui), who attest to the importance of the CZ concept. Both E-Surf reviewers suggested we improve the structure of the manuscript. As pointed out by Tunnicliffe, we now see the utility of bringing Section 3 forward in the article. This

will bring the discussion of the intellectual heritage ahead of the discussion of funding strategies to provide a more logical sequence that emphasizes the evolution of scientific thinking. We will also emphasize the four common elements of CZOs earlier in the paper, perhaps in the new section incorporating Section 3. We will also work on paragraphs 3 and 4 as suggested by Reviewer #2, so that they describe, in order, the research initiatives and evolution followed by the limits of those research programs and the potential to go beyond those achievements. As requested by Reviewer #2, we will also emphasize Tables 2, 3, and 4 in the manuscript.

Tunnicliffe points out that this re-organization could then allow the article to better live up to the title of the manuscript. We agree that perhaps the article as written and the title are a bit out of sync. After re-organizing and re-emphasizing, we will return to the title of the manuscript to see if a better title is warranted. Tunnicliffe also requests elimination of the use of the word "experiment" to describe the CZ science venture. We will edit out that word and use one of the more precise terms suggested.

Another set of words were also the focus of a few more of Tunnicliffe's comments – paradigm and transformative. "Paradigm" is defined by Merriam Webster as "a philosophical and theoretical framework of a scientific school or discipline within which theories, laws, and generalizations and the experiments performed in support of them are formulated" (accessed at https://www.merriam-webster.com/ on 9-7-17). Scientific paradigms include definitions of what should be studied, the questions of interest, and the broad approach of study. We argue that CZ science is at least a paradigm shift in that it emphasizes that the CZ is one entity and must be investigated in its entirety. Reviewer E. Bui agrees. Therefore, we propose to include a more complete discussion of why CZ science is a paradigm shift: but we will qualify our assertions appropriately. We will also plan to use the term paradigm only for the overall CZ science initiative and not for the emergent hypotheses in Section 6, and we will emphasize use of the word "transformative" for the CZ enterprise rather than the individual hypotheses in Table 4.

The reviewers had comments on figures that we will address. For example, Tunnicliffe

suggested changing Figure 1 to emphasize biological aspects. We will consider possible revisions for Figure 1. Given that both reviewers questioned Figure 3, we will modify the caption by pointing out in our revision that the figure includes sites associated as CZOs and that all the sites shown derive from networks within the U.S.A., Germany, France, and China, noting that some sites in China are co-funded and studied by scientists from the United Kingdom. The RBV and Critex networks (France) include sites outside of Europe. We will also add this information to the caption: RBV stands for the Réseau des Bassins Versants (Network of Drainage Basins), CRITEX is not an acronym, and TERENO stands for the Terrestrial Environmental Observatories.

Tunnicliffe and Bui made comments about CZ modelling efforts. Bui wrote that the future should emphasize CZ science that "facilitates extrapolation from CZO-based science and put[s] the results in broader regional and continental context. This means working with researchers . . . who build predictive spatial models of soil and geochemistry over large regions. Spatially distributed reactive transport models would be the ultimate objective here." She also argued for a future CZO science that ... "aims to predict the change trajectory that Earth systems might take under global warming (as proposed in the paper). This will require working with land surface modelers to refine the scale of their models." Likewise, Tunnicliffe wrote, "Table 3 seems to catalog a broad array of models, most of which were not specifically designed for CZ research. It would be more interesting to list emerging numerical models, or amalgamations of existing models, next to the specific CZ questions being pursued."

We agree with these comments about models completely. As Table 3 indicates, the initial CZ modeling efforts may be characterized into four groups. The first includes modifications and adoption of existing models to incorporate new couplings between hydrology and biogeochemistry, ecology and biogeochemistry, etc. The second includes identifying and filling critical gaps or knowledge of new processes such as hyporheic exchange, weathering, etc. The third includes development of a new generation of models that takes advantage of emerging streams of high resolution data such as airborne and UAV (unmanned aerial vehicle) based LiDAR and hyperspectral data. The fourth includes coupling between fast and slow processes across many time scales. Slow processes provide the template for the fast response variable, while the accumulative effect of the latter results in the evolution of the former. Both mathematical frameworks and data to support such modeling are still in their infancy.

Dialogue is ongoing as to whether the critical zone community will be best served through a single modeling framework or a library of existing models that allows more targeted exploration. The challenge lies in the central critical zone focus: ". . .generalizing and scaling place-based studies to principles-based understanding . . ." Place-based studies can demand very specific investigations that are highly tuned to the biogeomorphic setting of a specific location, but that provide little deeper understanding. In contrast, a model that is broadly applicable may simplify the representation of a given site so much that the model results in reduced accuracy of prediction. Therefore, both the advancement of critical zone science and critical zone modeling will likely progress in an intertwined manner. These issues will be articulated in the revised manuscript.

Tunnicliffe also writes, "Table 4 is missing any mention of hypotheses related to the social science aspects of the CZ. . .It would be good to see how this strand of the research fits in!" Likewise, reviewer E. Bui emphasizes that the future NSF network should "address current 'wicked' societal problems and help formulate better land development environmental management policies." We could not agree more. However, the CZO enterprise in the U.S.A. so far has not emphasized social science and no such hypothesis has yet emerged from the community. We will emphasize in the revised manuscript that such hypotheses are needed and should be part of the future of the network. For example, in the revision we plan to specifically mention the idea proposed by reviewers P. Shroeder and E. Bui that an urban CZO would be of great interest.

Reviewer #2 points out that "paragraph 7 mentions briefly the publication of numerous datasets (p. 7, l.1) sometimes spanning several decades of measurements. . .the

creation of this repository as well as the website . . .should be highlighted in the text."
We agree with the reviewer. The intent of the CZO network is to serve the research
community beyond those directly involved in the ongoing collection of CZ data for each
of the individual sites. As such, a key motivation of the network is the development of
publically available datasets pertaining to the structure and dynamics of the CZ under
investigation at each of the sites. The wide variety of CZO datasets can be accessed
through http://criticalzone.org/national/data/. As shown there, each of the CZOs are
collecting numerous data types. These datatypes commonly include sensor/sampler
network measurements showing time series response of different locations in the CZ to
meteoric events, spatially-resolved geophysical and geochemical measurements of CZ
structure, and LiDAR measurements of vegetation and bare earth topography, among
others. As discussed in the paper, there is a coordinated effort underway to ensure that
measurements are comparable across sites (i.e., the "common measurements" effort),
and that the posted datasets can be used by others to make cross-site comparisons
and conduct cross-site studies with existing data.

Tunnicliffe noted that "Table 2 does not back up your point about long-term measure-
ments. It would be more helpful to see the length of these records, rather than a smat-
tering of similar measurements that may or may not relate to broader hypotheses being
tested across CZOs." This reviewer also noted that, "Pg 9, ln 307 makes reference to
the 'extremely long' duration of the datasets - this could use some quantification."

We will clarify these points in the revision. In short, the time-series datasets (sensor
and sampler arrays, eddy covariance, hydrometeorology, vadose zone and saturated
zone aqueous chemistry, etc.) have durations that are roughly equivalent to the age of
the CZO sites, determined by the initiation of NSF funding, with the caveat that CZOs
have often added new study locations that were not among the original set. Three
sites (SSCZO, BCCZO and SSHCZO) have been in operation since 2007, and so their
longer-term observational datasets extend roughly over that duration. Three other
sites (CJCZO, LQCZO and CRCZO) that initiated operations two years later have

measurements dating to 2009, and three newer sites (IMLCZO, CHCZO, RCCZO and ERCZO) have datasets dating to 2013. Therefore, at present, continuous time series datasets range in duration from ca. 4 to 10 years. In addition, however, several of the sites are located in sites that provide longer datasets through previous measurement programs. The question of duration of dataset is thus somewhat complex, but we will try to make this information more transparent in the revision.

Please also note the supplement to this comment:
https://www.earth-surf-dynam-discuss.net/esurf-2017-36/esurf-2017-36-AC1-supplement.pdf

---

## Author Response (AR1)

Authors' response: Esurf-2017-36

We appreciate the comments on our paper reviewing the arc of CZ research and proposing a strategy for the future design of an observatory network. We found the reviewers' comments thoughtful and helpful as discussed below. We also appreciated the comments from outside the U.S.A., specifically reviewers from Sweden (K. Bishop) and Australia (E. Bui), who attest to the importance of the CZ concept.

Both E-Surf reviewers suggested we improve the structure of the manuscript. As pointed out by Tunnicliffe, we brought Section 3 forward in the article. This brings the discussion of the intellectual heritage ahead of the discussion of funding strategies to provide a more logical sequence that emphasizes the evolution of scientific thinking. We also now emphasize the four common elements of CZOs earlier in the paper in the new section incorporating Section 3. We have also revised paragraphs 3 and 4 as suggested by Reviewer #2, so that they describe, in order, the research initiatives and evolution followed by the limits of those research programs and the potential to go beyond those achievements. As requested by Reviewer #2, we also call out Tables 2, 3, and 4 in the manuscript.

Tunnicliffe points out that this re-organization allows the article to better live up to its title. We agree that perhaps the article as initially written and the title were a bit out of sync. After re-organizing and re-emphasizing, we returned to the title of the manuscript to see if a different title was warranted and we decided to keep the title. Tunnicliffe also requested elimination of the use of the word "experiment" to describe the CZ science venture. We have edited out that word and now used a more precise term (enterprise), as suggested.

Another set of words were also the focus of a few more of Tunnicliffe's comments – paradigm and transformative. "Paradigm" is defined by Merriam Webster as "a philosophical and theoretical framework of a scientific school or discipline within which theories, laws, and generalizations and the experiments performed in support of them are formulated" (accessed at https://www.merriam-webster.com/ on 9-7-17). Scientific paradigms include definitions of what should be studied, the questions of interest, and the broad approach of study. We argue that CZ science is at least a paradigm shift in that it emphasizes that the CZ is one entity and must be investigated in its entirety. Reviewer E. Bui agrees. Therefore, we have included a more complete discussion of why CZ science is a paradigm shift, and we qualify our assertions appropriately. We do respond to Tunnicliffe by using the term paradigm only for the overall CZ science initiative and not for the emergent hypotheses in Section 6, and we emphasize use of the word "transformative" for the CZ enterprise rather than the individual hypotheses in Table 4.

The reviewers had comments on figures that have now been addressed. For example, Tunnicliffe suggested changing Figure 1 to emphasize biological aspects. We have considered possible revisions for Figure 1 and have selected one to replace the earlier version of that figure, because it more clearly integrates biology with the weathering profile and it has been commonly used by the CZ science community to highlight these interactions. Given that both reviewers questioned Figure 3, we have modified the caption by pointing out in our revision that the figure includes sites associated as CZOs and that all the sites shown derive from networks within the U.S.A., Germany, France, and China, noting that some sites in China are co-funded and studied by scientists from the United Kingdom. The sites included are also ones that are registered on SiteSeeker. The RBV and Critex networks (France) include sites outside of Europe. We added this information to the caption: RBV stands for the Réseau des Bassins Versants (Network of Drainage Basins), CRITEX is not an acronym, and TERENO stands for the Terrestrial Environmental Observatories.

We agree completely with the comments about models by Tunnicliffe and Bui. As Table 3 indicates, the initial CZ modeling efforts may be characterized into four groups. The first includes modifications and adoption of existing models to incorporate new couplings between hydrology and biogeochemistry, ecology and biogeochemistry, etc. The second includes identifying and filling critical gaps or knowledge of new processes such as hyporheic exchange, weathering, etc. The third includes development of a new generation of models that takes advantage of emerging streams of high resolution data such as airborne and UAV (unmanned aerial vehicle) based LiDAR and hyperspectral data. The fourth includes coupling between fast and slow processes across many time scales. Slow processes provide the template for the fast response variable, while the accumulative effect of the latter results in the evolution of the former. Both mathematical frameworks and data to support such modeling are still in their infancy. We discuss this in the paper but we do not separate the table explicitly because of the complexity of real distinctions along these lines for many of the models.

Tunnicliffe also writes, "Table 4 is missing any mention of hypotheses related to the social science aspects of the CZ…It would be good to see how this strand of the research fits in!" Likewise, reviewer E. Bui emphasizes that the future NSF network should "address current 'wicked' societal problems and help formulate better land development environmental management policies." We could not agree more. However, the CZO enterprise in the U.S.A. so far has not emphasized social science and no such hypothesis has yet emerged from the community. We emphasize in the revised manuscript that such hypotheses are needed and should be part of the future of the network. For example, in the revision we specifically mention the idea proposed by reviewers P. Shroeder and E. Bui that an urban CZO would be of great interest.

Reviewer #2 points out that "paragraph 7 mentions briefly the publication of numerous datasets (p. 7, l.1) sometimes spanning several decades of measurements…the creation of this repository as well as the website …should be highlighted in the text." We agree with the reviewer. Tunnicliffe noted that "Table 2 does not back up your point about long-term measurements. It would be more helpful to see the length of these records, rather than a smattering of similar measurements that may or may not relate to broader hypotheses being tested across CZOs." This reviewer also noted that, "Pg 9, ln 307 makes reference to the 'extremely long' duration of the datasets - this could use some quantification."

We have clarified these points in the revision.  In short, the time-series datasets (sensor and sampler arrays, eddy covariance, hydrometeorology, vadose zone and saturated zone aqueous chemistry, etc.) have durations that are roughly equivalent to the age of the CZO sites, determined by the initiation of NSF funding, with the caveat that CZOs have often added new study locations that were not among the original set.  Three sites (SSCZO, BCCZO and SSHCZO) have been in operation since 2007, and so their longer-term observational datasets extend roughly over that duration.  Three other sites (CJCZO, LQCZO and CRCZO) that initiated operations two years later have measurements dating to 2009, and four newer sites (IMLCZO, CHCZO, RCCZO and ERCZO) have datasets dating to 2013.  Therefore, at present, continuous time series datasets range in duration from ca. 4 to 10 years. In addition, however, several of the CZOs are located in sites that provide longer datasets through previous measurement programs.  The question of duration of dataset is thus somewhat complex, but we have tried to make this information more transparent in the revision.

[revised manuscript text omitted]

developed often was forced to use had to rely on statistical approaches to explain variability
instead of developing more fundamental explanations that was caused by based on underlying
geological heterogeneity and its origins. Recognizing this the need to emphasize the geological
underpinnings of place-based science in the late 2000s, researchers within the water, soil,
geochemistry, and geomorphology communities began articulating a need for integrated science
across the entire zone from canopy to bedrock to incorporate the full significance of the
underlying geology (Anderson, 2004; Brantley et al., 2006; Chorover et al., 2007; U.S.
Committee on Integrated Observations for Hydrologic and Related Sciences, 2008; U.S. Steering
Committee for Frontiers in Soil Science, 2009; U.S. National Research Council, 2010; Banwart
et al., 2011; Committee on New Research Opportunities in the Earth Sciences at the National
Science Foundation, 2012; White and Sharkey, 2016). These researchers also recognized that
advances in studying Earth's surface were fragmented precisely because investigators typically
targeted many different sites without coordination among disciplinary approaches. Under the
paradigm of small funded projects, different types of measurements were made but they were
completed at so many different sites that integration of observations into models was difficult to
impossible.

Eventually the need to study the CZ as one integrated entity resulted in the NSF program
establishing the Critical Zone Observatory program in 2007 (White et al., 2015). In this
initial phase, three CZOs were funded (Anderson et al., 2008). Two years later, three
more CZOs were funded. By 2013 this number had grown to nine observatories supported
through a competitive selection process. In addition to the expansion of sites in 2013, a CZO
National Office (NO) was established by NSF in 2014 through a competitive process,
with the intent of providing the CZO Network with an administrative
structure for furthering coordination (White et al., 2015). The
number of CZOs has remained stable through 2017.

Inauguration of the CZO program implicitly defined the term "critical zone observatory"
to be distinct within the long history of observatories in the US and abroad as an observatory that
promotes study of the entire CZ as one entity. As implemented today, CZOs are sites or closely
connected sets of sites with no required size or specified range of conditions.  In fact, the
physical scope of a CZO is set only by the fundamental questions driving the establishment of
the observatory.  A fundamental characteristic of a CZO is that it is able to operate over a long
enough period to quantify controlling mechanisms thoroughly and to capture
temporal trends that reveal how the critical zone operates.  Two more
characteristic of a CZO are that it is amenable to study by many disciplines and that it
integrates understanding of long- and short-timescale phenomena.  Finally, each
CZO operates as an adaptive and agile hypothesis-testing machine, not simply a monitoring
program.

An  CZOs
evolved in the U.S., they began to play nine important
emergent roles within the environmental scientific endeavor. These are
described in the next section.

**5 The nine emergent roles of CZOs**

Here we highlight the nine important roles that can only be accomplished at of an observatory are described and amplify the description of each role with examples of scientific results from across the CZO network today.

First, CZOs act as *synthesizers of interdisciplinary research into convergent approaches* at one specific site that lead to emergent novel understanding and ultimately result in more deeply-informed process-based modelsgeneralized and predictive understanding (e.g.Rasmussen et al., 2011b; Kumar et al., 2017, in review) (Rasmussen et al., 2011a). In other words, observatories induce scientists from different disciplines to make measurements using different disciplinary approaches at the same location instead of making them at disparate sites, driving cross-disciplinary understanding that crosses the boundaries of disciplines in describing CZ function (Hynek et al., 2016; Sullivan et al., 2016; Yan et al., 2017; Chen et al., 2017 in press). To dateAt first, much of the synthesis has crossed only two disciplines at a time: for example, several papers have emphasized how geomorphological concepts related to erosion must be incorporated to understand chemical weathering, and vice versa (Rempe and Dietrich, 2014; Riebe et al., 2016). Likewise, researchers have related tree roots to water cycling (Vrettas and Fung, 2015). Now, researchers are targeting But many puzzles remain concerning multi-disciplinary aspects of CZ entities, and much effort is currently focusing on such frontiers. For example, at the Calhoun CZO, where the South Carolina landscape was severely eroded by cotton farming, logistic regression models treat market and policy conditions in the context of topographic characteristics (Coughlan et al. 2017). In another example, distributed CZO data were used to develop a predictive framework linking regolith structure, root-zone storage and vegetation to dry periods (Klos et al., 2017, in review). CZOs haveBy fostered fostering measurements from all disciplines in centralized places and this is the first step toward addressing broader multi-disciplinary models describing CZ function, CZOs are discovering not only how to cross disciplines but how individual such disciplines can converge(Brantley et al., 2017, in press; Holbrook et al., 2017, in review).

Second, CZOs provide *stable platforms for long-term measurements made over long-term durations* (Table 2). Some of the datasets synthesized by CZOs are now available at CZOs are extremely longdecadal to multi-decadal in durationfor decades or several decades, enabling the testing of effects of inter-annual climatic variation effects as well as longer-term climate or land-use trends. For example, the Reynolds Creek CZO recently published 31 years of hourly data that is are spatially distributed at 10 m resolution for air temperature, humidity, and precipitation amount and phase across the 239 km$^2$ Reynolds Creek Experimental Watershed (Kormos et al., 2016) and a . Reynolds also published another 10-year data set that spans the rain-snow transition zone (Enslin et al., 2016). Similarly, using water balance, climate and satellite datasets, Zapata-Rios et al. (2016) demonstrated decreasing trends in water and energy influx in the Jemez CZO to the CZ over the past 30 years were in the Jemez CZO and recently related to discussed the implications of these long-term trends for CZ structure
development(Zapata-Rios et al., 2016). Major changes in soil biogeochemistry have been
documented by Calhoun CZO researchers over 50-years of reforestation in fields cultivated for
cotton (Mobley et al., 2015). T as inferred from present day CZ structural datasetshat CZO also
spearheads an effort to recover archived data from three eroded watersheds that were
farmed from the late 1940s to 1962 – as well as to re-instrument the catchments. -Many
other multi-yearof the long-term datasets made at CZOs are measurements that are common to
all CZOs and enable have been designed to test the hypothesis that new understanding will
hypothesis-testingemerge about processes and function from such cross-cutting data. For
example, characterization of dissolved organic matter (DOM) coordinated measured with similar
methodology across five CZOs revealed a strong role for CZ structure in setting the origin,
composition and fate of DOM in streams (Miller et al., 2016). In another example, a coordinated
effort emerged to measure and understand the relationships among chemical species'solute
concentrations and water discharge in streams (e.g. Kirchner, 2003; Godsey et al., 2009). This
led to papers published by CZ scientists in aA special issue on the topic of *Water Resources*
*Research* (Chorover et al., 2017, in press) is pointing the way toward the use of . The goal now is
to use CZO knowledge of subsurface structure to explain concentration-discharge behavior *a*
*priori* for other settings.

Third, CZOs act as *a stimulus and test-bed for modelling and prediction.* Modeling the
CZ poses ais a unique challenge in that models must of-address ing the coupling across a very
broad range of time scales from seconds to millennia weather driven event dynamics to (Table
3). To tackle this challenge, CZOs are both adapting existing models and developing new
models. For example, one CZO is developing a hierarchy of modules to describe processes that
occur over seconds to millennia (Duffy et al., 2014). millennium scale geologic formation. This
intertwining of functional outcome at the short time scale resulting from the form or structure
formed at the long time scale, within the context of specific hypothesis creates unique challenges
of coupling across many time scales within a modeling framework. At the same time, availability
of novel measurements in space, time and depth is enabling CZOs to ask new questions that were
not possible before. To address these challenges and opportunities, CZOs have been adapting
existing models or developing new ones to answer questions that seem pertinent to a specific

[revised manuscript text omitted]

communicate as effectively, for example, on about sapflow in trees as about on seasonal
variations in groundwater flow, as the two are increasingly recognized as interdependent.
Collaborations are constantly developing and allowing scientists to see problems in new ways,
and accepting new and with different perspectives. Ideas are growing across the network about
regolith formation (Riebe et al., 2016), snow hydrology (Harpold et al., 2014; Tennant et al.,
2017), microbial diversity (Fierer et al., 2003), trees (Brantley et al., 2017, in press), critical
transitions (Kumar et al., 2017, in review), and many other topics. We have produced an
enormous datasets (http://criticalzone.org/national/data/datasets/). We no longer treat parts of the
CZ as mereisolated components or black boxes: instead, we incorporate more specificity and
understanding when we describe the integrated system. We are finding innovative ways to
communicate the CZ concept to the public. We have stimulated and promoted development of
CZOs worldwide.

Although there have been many successes, we have also observed several weaknesses in
the approach as it is constituted today. Since each observatory is individually funded based on
the merits of its targeted science and management, there is an obvious competition within each
CZO for allocation of resources in addressingto address common (network-wide) measurements

[revised manuscript text omitted]

Formatted

Formatted Table

Formatted

Formatted

Formatted

Formatted

Formatted

Formatted

Formatted

Formatted

Formatted

Formatted

Formatted

Formatted

Formatted

Formatted

Formatted

Formatted

Formatted

Formatted

Formatted

Formatted

Formatted

Formatted

Formatted

Formatted

Formatted

Formatted

Formatted

Formatted

Formatted

Formatted

Formatted

Formatted

Formatted

Formatted

Formatted

| | | | |
|---|---|---|---|
| AWESOM | Atmosphere/, Watershedwatershed/, Ecologyecology/, Stream streamand /Ocean ocean Model | Possiblex | 
[revised manuscript text omitted]